# Decline in Other Instrumental Activities of Daily Living as Indicators of Driving Risk in Older Adults at an Academic Memory Clinic

**DOI:** 10.3390/geriatrics8010007

**Published:** 2023-01-05

**Authors:** Frank Knoefel, Shehreen Hossain, Amy T. Hsu

**Affiliations:** 1Bruyère Research Institute, Ottawa, ON K1N 5C8, Canada; 2Bruyère Continuing Care, Ottawa, ON K1N 5C8, Canada; 3Faculty of Medicine, University of Ottawa, Ottawa, ON K1H 8L6, Canada; 4Department of Systems and Computer Engineering, Faculty of Engineering and Design, Carleton University, Ottawa, ON K1S 5B6, Canada; 5AGE-WELL NIH—SAM3, Ottawa, ON K1N 5C8, Canada; 6Ottawa Hospital Research Institute, The Ottawa Hospital, Ottawa, ON K1Y 4E9, Canada

**Keywords:** driving cessation, cognitive decline, quantitative data, logistic regression, function

## Abstract

Background: Decisions around driving retirement are difficult for older persons living with cognitive decline and their caregivers. In many jurisdictions, physicians are responsible for notifying authorities of driving risks. However, there are no standardized guidelines for this assessment. Having access to a driving risk assessment tool could help older adults and their caregivers prepare for discussions around driving retirement. This study compares the clinical profiles of older adult drivers assessed in an academic memory clinic who were referred to the driving authority to older drivers who were not with a focus on instrumental activities of daily living (iADLs). Methods: Data on referred (R) and not-referred (NR) drivers were extracted from medical records. Elements from the medical history, cognitive history, functional abilities, Modified Mini-Mental State (3MS) examination, Trails A/B, and clock drawing were included in the analysis. Four risk factors of interest were examined in separate logistic regression analyses, adjusted for demographic variables. Results: 50 participants were identified in each group. The R group was older on average than the NR. As expected, R were more likely to have Trails B scores over 3 min and have significantly abnormal clock drawing tests. R also showed lower 3MS scores and a higher average number of functional impairments (including managing appointments, medications, bills, or the television). Conclusion: Beyond standard cognitive tests, impairment in iADLs may help general practitioners identify at-risk drivers in the absence of standardized guidelines and tools. This finding can also inform the design of a risk assessment tool for driving and could help with approaches for drivers with otherwise borderline test results.

## 1. Introduction

According to Statistics Canada, in 2009 there were over 3 million Canadians 65 years and older who had a valid driver’s license, with that number expected to double by 2050. In addition, among the 28% of older Canadians diagnosed with dementia, some 20,000 people had a driver’s license [1]. There is abundant literature about the importance of driving to older adults, especially regarding their independence. Retirement from driving can lead to depression [2,3], isolation [4], cognitive decline [5,6], and overall decline in function, potentially leading to admission to long-term institutional care [6].

Driving is a complex instrumental activity of daily living (iADL). There is extensive literature on the cognitive domains that are associated with driving risk; these include visual attention [7], visuospatial ability [8], speed of processing [9] and executive function [10,11,12]. Work examining executive functioning suggests that it is important in novel and demanding situations requiring rapid adaptation as compared to usual dominant responses [13]. Like most iADLs, the ability to drive is affected by changes in cognition and physical health [14]. The single biggest risk factor for decline in cognition and dementia is age [15]. The prefrontal cortex is associated with executive functioning [8] and seems to be particularly affected by aging [16]. There is some radiologic evidence that changes in the frontal gray matter can impact driving risk [17].

In many jurisdictions around the world, physicians are required to notify driving authorities regarding medical-related driving risks in their patients. There is literature that shows how traumatic losing a driver’s license is for older adults, leading to reluctance and a sense of loss [18,19]. Some drivers react with shock and anger [20]. Overall, the literature suggests that older adults and their caregivers do not sufficiently prepare for driving retirement [21,22]. In fact, it has been suggested that formal written forms be used to facilitate the conversation between clinicians and older adults regarding driving retirement, with one proposed model being “Advance Driving Directives” [22]. However, there is a gap in the literature on tools that older adults and their caregivers can use to self-assess driving risk. The lack of standardized guidelines and tools is also problematic for primary care providers who may be less familiar with performing neurological exams and cognitive assessments. As most primary care physicians may not have access to or time for the Trails B, clock drawing, or full cognitive history of the patient, other indicators like difficulties with iADLs may be informative in their assessment of driving risk. In fact, a significant relationship was found between the Assessment of Motor and Process Skills (AMPS) iADL tool and driving performance [23]. However, there have been few studies examining the association between iADL impairment and driving risk. 

The objective of this exploratory study was to compare the clinical profiles of older adult drivers assessed in an academic memory clinic who were referred to the driving authority to older drivers who were not, with a focus on iADLs–something that older adults and their caregivers, as well as primary care physicians can evaluate without having to use clinical tools. The long-term goal is for this study to inform the development of a risk assessment tool that older adults and their care partners could refer to on-line to help them prepare for the challenges of driving assessments to come. 

## 2. Data Collection and Analysis

### 2.1. The Setting

The Bruyère Memory Clinic is the only academic clinic dedicated exclusively to cognitive assessment in Ottawa, Ontario, Canada, a city of just over 1 million inhabitants. The population is made up of a majority of English- and French-speaking Caucasians with a smaller visible minority. Clinical assessments are provided by 2 cognitive neurologists, 2 geriatricians and 2 family physicians with additional training in care of the elderly. 

The usual clinic process begins with a community referral. Patients are asked to bring a completed questionnaire regarding medical history to their first appointment with a nurse. The nurses, who are trained to administer the tests by the neuropsychologists, then administer the Modified Mini-Mental State Examination (3MS: [24]), the Boston Naming Test (short form: [25]), the Trails A and B [26], and the clock drawing test. Testing is done in either French or English, the two official languages in Canada. Test versions were validated in both languages. The clock drawing is scored from 1 to 4, with 1 having all numbers in approximately the correct place and the hands correctly indicating “10 past 11”, 2 having a mistake with hand lengths, 3 having the hands pointing to the wrong numbers, and 4 having more complex errors than these (e.g., numbers in the wrong places, too many numbers, no numbers, no hands). 

At the appointment that follows, physicians complete the medical history, review of medication, and take a detailed cognitive and functional history. They then perform a detailed neurological exam and may repeat parts of the cognitive assessment. Following this, a diagnostic impression is presented and driving risk is discussed. In follow-up appointments, physicians typically review symptoms and repeat some sort of cognitive testing and continue to review driving safety. For expedience and sensitivity for mild cognitive impairment, some physicians use the Montreal Cognitive Assessment (MoCA: [27]) at follow-up appointments. At any point in this process, the physicians have the option to refer to one of 3 neuropsychologists for more extensive testing.

The usual practice is that physicians use the Trail B, clock drawing, and history as the key determining factors for driving risk. Overall cognitive ability is also considered, but typically to a lesser extent. In Ontario, physicians and other clinicians are legally required to report adults they suspect of having a high driving risk. If high driving risk is determined, and after discussion with the patient and family, the physician is required to write a letter to the Ministry of Transportation of Ontario (MTO).

### 2.2. The Participants

The first step was the identification of 50 consecutive referrals to the driving authority. Our data extraction began in November 2021, and letters that were sent to the MTO were reviewed in reverse chronological order. Patients were included in the study if they had a relatively complete data set–none of the key documents could be missing, but one or two elements of a document (e.g., 3MS) could be. Another inclusion criterion was that the participant was required to have had a 3MS, a clock drawing and Trails A and B completed within the last 6 months of the appointment where the letter decision was made. The included older adults comprised of individuals with any of the following diagnoses: normal aging, subjective cognitive decline, mild cognitive impairment or early to mild dementia. Patients with active psychiatric conditions were excluded, however some patients were mildly depressed and could have been taking anti-depressants.

A similar approach was used to identify the comparison group. Starting in November 2021 and working backwards, 50 consecutive patients who were drivers and were not referred to the MTO were identified. Again, to be included in the study they needed to have all the key elements available, with allowance for a couple of missing data points. The last 3MS, clock drawing and Trails A and B had to have been "completed" within 6 months.

### 2.3. The Data

The data used in this study is clinical data, meaning this is secondary use of the data. The items selected for analysis were ones that were felt to be documented with some consistency and with a minimum of subjectivity. The following demographic items were therefore included in the analysis: age, sex, education, first language, and living status (alone, with other). The following items were extracted from the history: cardiovascular diagnosis, number of medications, and psychiatric diagnosis (yes/no). The cognitive history included the presence of symptoms in the following domains: memory, orientation to place, language, and executive functioning. The functional history included decline in the ability to manage appointments, medication, bills, a computer, and the TV remote. The 3MS total score was included, as were the sub-scores. An adjusted 3MS total score was utilized for the analysis for 9 patients, who were assessed and given a total potential score by one of the physicians at the clinic (FK) to account for incompleteness (i.e., some sub-scores were missing). This was done, for instance, when the test was administered by telephone and drawing of intersecting pentagons and/or the three-step command could not be completed. The clock drawing score was included as were Trails A and B (in seconds). For the regression model, Trails B was dichotomized as normal (0–119 s) to moderate impairment (120–179 s) versus impaired (180 s and over).

### 2.4. Data Analysis

We examined the distribution of clinical profiles among older adult drivers, stratifying the results by whether the patients were referred to the driving authority (Ministry of Transport of Ontario, MTO). Given our small sample size, we have used Fisher’s exact test to compare all categorical variables and t-test to compare scale and other continuous variables. Variables were assessed for collinearity by Pearson’s and Spearman’s correlation coefficient. No correlations exceeded 0.70. Logistic regression models were fitted to estimate odds ratios for variables predicting the risk of being referred to the MTO. Two-tailed *p*-values of 0.05 or less were considered statistically significant. SAS 9.4 was used to perform all statistical analyses. 

## 3. Results

Our participants consisted of 50 older adult drivers who were referred to the MTO and 50 older adult drivers who were not referred to the MTO. Baseline characteristics are listed in Table 1 by their referral status. Adult drivers referred to MTO were older (78.2 ± 6.6 vs. 72.9 ± 7.0 years, *p* < 0.05) and were less likely to have a college or university degree (54% vs. 72%, *p* = 0.10) than non-referred drivers. Compared to non-referred drivers; referred drivers were more commonly male (62% vs. 50%, *p* = 0.31), spoke English as their first language (50% vs. 34%, *p* = 0.08) and lived alone (22% vs. 12%, *p* = 0.29). Referred drivers reported a slightly higher average number of medications (4.3 ± 2.7 vs. 4.2 ± 2.3, *p* = 0.72) and reported having a psychiatric condition (36% vs. 34%, *p* = 1.00) more than non-referred drivers. 

Table 2 shows the distribution of participants’ clinical risk factors for driving, stratified by referral status. Referred drivers showed a significantly higher average number of domains with functional decline (1.96 ± 1.5 vs. 0.72 ± 1.0, *p* < 0.05) compared to non-referred drivers. About half of the referred drivers demonstrated functional declines in managing iADLs, including appointments (50% vs. 24%, *p* < 0.05), medications (52% vs. 18%, *p* < 0.05) or bills (44% vs. 16%, *p* < 0.05). Two-thirds of the referred drivers exhibited high impairment (70% vs. 6%, *p* < 0.05) on the Trails B test. Their 3MS total scores were also significantly lower (76.46 ± 15 vs. 91.2 ± 7.7, *p* < 0.05). Results on the 3MS sub-scores can be found in Appendix A. Almost half of the referred drivers (47% vs. 8%, *p* < 0.05) demonstrated poorer performance in the clock drawing test, including having the clock hands pointing to the wrong numbers, or having more complex errors (e.g., numbers in the wrong places, too many numbers, no numbers, no hands). 

Table 3 shows the extent to which functional decline in managing different iADLs independently influenced the odds of being referred to the MTO, after adjusting for age and education. Results from the logistic regressions suggest that an increase in the count of functional impairments (OR 2.01; 95% CI 1.4–2.9; *p* < 0.05), was significantly associated with being referred to the MTO. Separately, each iADL was also significantly and independently associated with being referred. Specifically, the odds of a referral was 2.64 (95% CI 1.1–6.6; *p* < 0.05) in older adults who had declined in their ability to manage appointments, 5.64 (95% CI 2.0–15.6; *p* < 0.05) in those who declined in their ability to manage medications, 4.43 (95% CI 1.6–12.3; *p* < 0.05) with those having more difficulty managing bills, and 12.6 (95% CI 1.5–105.1; *p* < 0.05) in those with more difficulty operating the TV.

Figure 1 illustrates the predicted probability that an older adult was referred to the driving authority, based on categories of the clinical risk factors. Estimates can also be found in Appendix A. With an increase in the number of iADL domains showing decline, the mean predicted probability of getting a referral is higher in referred drivers’ group (range: 0.37–0.86) compared to non-referred drivers (range: 0.28–0.69). 

Upon further analysis of the typology of iADL decline (Table 4), it can be seen that not having other iADL decline (iADL = 0) was more common in the non-referred group (60%) than the referred group (28%). Comparing referred to non-referred drivers, the distributions of iADL declines were evenly spread across the two groups in individuals with a decline in a single domain and in those with declines in two iADLs. However, referred drivers were over-represented among individuals with declines in three to four iADLs. The most commonly observed deficits were in managing medications and bills, followed by appointments, then in TV and computer especially among the referred group. Compared to those with at least two iADL impairments in the referred group (Appendix A), those with two iADL impairments in the non-referred group were younger (non-referred: 72.9 ± 7.0 years, referred: 78.9 ± 5.3 years), more educated (non-referred: 76.9% vs. referred: 46.9% who attended college or university), and more likely to live with others (non-referred: 92.5%, referred: 75.0%). Furthermore, the non-referred group performed better on the Trails B (non-referred: 102.7 ± 27.7 s, referred: 219.8 ± 85.2 s) and clock drawing tests (see Appendix A). It is interesting to note that there were three participants with 3 iADL declines who were not referred to the driving authority: their ages were all below 75, their 3MS scores were above 80, their Trail B scores were below 150 s, and their clock drawings did not show difficulty with placing hands on the correct numbers.

## 4. Discussion

In this study, we compared the clinical profiles and driving risk among older adult drivers who were assessed at an academic memory clinic with the purpose of identifying elements that older adults and their caregivers could use to help prepare for medical appointments where driving might be discussed. The findings of Trails B and clock drawing triggering referral to the driving authority were expected as that is the clinical practice, and the results align with previous work [28,29]. Similarly, it is not surprising that lower scores on the 3MS, suggesting increasing cognitive decline, also predicted referral to the driving authority. However, these tests are not meant to be self-administered, and would not help older adults and their caregivers assess driving risk. On the other hand, iADLs are something that can be self-assessed and were found to have a statistically significant association with a referral. 

This study found that with every additional iADL where an older adult had difficulty, there was an additional 2-fold increase in relative risk to being referred to the driving authority. In the context of driving being a complex iADL [23], and the previous work using the AMP [23], it is not surprising that impairment in more iADLs is associated with a greater chance of referral. The iADLs used in this study can be divided into 2 major groups: those that require complex interactions between multiple cognitive domains: managing appointments, managing medication, and paying bills; and those that are more related to managing complex devices: managing a television remote and managing a computer. Both of these types of iADLs would have important overlaps with driving ability. As an example, taking medication requires some understanding of what the medication is for (memory) and a good sense of time (orientation) to ensure the correct medications are taken at the correct time. In addition, this requires planning (e.g., taking medication when away from home), execution (i.e., taking the medication at the correct time), and reassessment/correction if a mistake was made. This aligns with some of the cognitive requirements for driving. First, it requires having an overall knowledge of how a vehicle functions. Similarly, having a sense of time is essential for both the tactical (safe merging into traffic, safe stopping) and strategic (how much is required for this trip) sides of driving. Finally, the driver needs to continuously adapt to the results of their driving with respect to the traffic and road conditions and the actions of other drivers. Managing a complex device, like a computer, requires understanding how the device works and the ability to learn to use the software optimally (memory). Again, executive functioning is required to plan, execute, and reassess/correct. Using a computer optimally means using the best available software for the job. One might try one software but then try another and compare results. Similarly, a vehicle is a complex device with various parts that need to be understood. There are many ways to slow the vehicle: foot off the gas, foot on the brake, using the parking brake, and while driving any combination of these may create the best results. Given this, having difficulty with one or more of these iADLs should affect driving ability. 

Surprisingly, three participants with 3 iADL declines were not referred to the driving authority. None of these were considered high risk at the time of assessment because of their clock drawings and Trail B scores. However, all three were flagged for driving risk reassessment. This group was more educated and more likely to live with others than the referred group, which may have affected the decision by the physician. It could also be that these participants were at higher driving risk than their cognitive testing suggested. It could also be that a formal iADL tool, like the AMPS, would have changed the decision to refer. More work needs to be done to clarify this. 

The utility of iADL impairments in assessing an older adult’s driving risk has not been well-studied in the extant literature. Findings from our analysis suggest this is a promising indicator that can be used by primary care physicians and other clinicians who are tasked with assessing an older patient’s driving risk and when making a determination for a referral to the MTO. 

### Limitations

This study is limited by the small sample size and the data being from a single clinic, which may affect the generalizability of these findings. Data on ethnicity was not collected, so it is not possible to determine its role on referral patterns. It is important to recall that the outcome in this study was referral to driving authority by a specialist and not the results of an on-road driving test. There is also the possibility that the clinicians in this study showed other biases, for instance, age that was identified as showing a correlation.

Despite these limitations, we were able to demonstrate a strong relationship between challenges with iADLs and referral to the driving authority. Future work should include a larger sample size including multiple sites. Another weakness is that the list of iADLs used was extracted from the charts as free text. Future work could look at using a formal method to assess iADLs. Future research could then see if there was a sequencing of iADL decline or different combinations of impaired iADLs that were more likely to predict driving risk. Furthermore, the association between iADLs and driving risk could be confirmed by driving studies, be they on-road tests, naturalistic driving or driving simulation.

The information generated from this work could contribute to a self-assessment tool used by older adults and their families but might also be useful for clinicians who do not have access to more detailed neuro-cognitive assessments or when their patients have borderline scores on cognitive testing. 

## 5. Conclusions

This study explored the relationship between iADLs and the risk of getting referred to a driving authority at an academic memory clinic. While most memory clinics have the resources to do detailed neuro-cognitive testing, primary care physicians are typically the first to see older adults with potential cognitive decline. This latter group may not have the skills or time to do more complete neuro-cognitive assessments. The more tools they have in available to them, the more likely the correct patients will be referred to driving authorities. This paper suggests that having more iADL decline is related to having increased driving risk as assessed in an academic memory clinic. Future work will hopefully result in prediction models that older adults and their caregivers can use to facilitate the discussion of driving retirement in the case of older adults with cognitive decline. Ultimately, if older adults can do a self-assessment, which might include iADLs, prior to their appointment with their physician, it will be easier for all involved to have the difficult conversation regarding driver retirement.

## Figures and Tables

**Figure 1 geriatrics-08-00007-f001:**
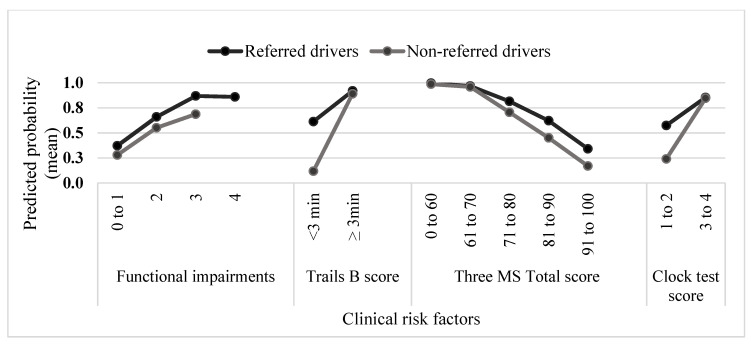
Mean predicted probability of being referred to driving authority for different clinical risk factors.

**Table 1 geriatrics-08-00007-t001:** Baseline characteristics by referral status.

	Referred Drivers (N = 50)	Non-Referred Drivers (N = 50)	*p*-Value
Age (years), mean ± SD	78.2 (6.6)	72.9 (7.0)	<0.05
Sex, n (%)			
Male	31 (62)	25 (50)	0.31
Female	19 (38)	25 (50)	
Education, n (%)			
High school or lower	23 (46)	14 (28)	0.10
College or University	27 (54)	36 (72)	
First language, n (%)			
English	25 (50)	17 (34)	0.08
French	25 (50)	30 (60)	
Other	0 (0)	3 (6)	
Living status, n (%)			
With other	39 (78)	44 (88)	0.29
Alone	11 (22)	6 (12)	
Cardiovascular disease, n (%)			
Yes	38 (76)	41 (82)	0.62
No	12 (24)	9 (18)	
Psychiatric disease, n (%)			
Yes	18 (36)	17 (34)	1.00
No	32 (64)	33 (66)	
Number of medications, mean ± SD	4.3 (2.7)	4.2 (2.3)	0.72

**Table 2 geriatrics-08-00007-t002:** Description of clinical risk factors for driving by referral status.

Clinical Risk Factor	Referred Drivers (N = 50)		Non-Referred Drivers (N = 50)		*p*-Value
		N Missing		N Missing	
Functional impairments, mean ± SD	1.96 ± 1.5	0	0.72 ± 1.0	0	<0.05
3MS Total score, mean ± SD	76.5 ± 15.0	0	91.2 ± 7.7	0	<0.05
Trails B test (seconds), mean ± SD	236.5 ± 77.3	3	119.6 ± 45.3	1	<0.05
Trails B score categories, n (%)					
Below 3 min	14 (30)	3	46 (94)	1	<0.05
3 min or above	33 (70)		3 (6)		
Clock drawing score categories, n (%)					
Score 1–2	26 (53)	1	46 (92)	0	<0.05
Score 3–4	23 (47)		4 (8)		

**Table 3 geriatrics-08-00007-t003:** Associations between iADL impairment and being referred to the driving authority.

iADL Functional Impairments	Adjusted OR (95% CI)	*p*-Value
Count of iADLs with decline	2.01 (1.4–2.9)	<0.05
Decline in managing specific iADL domains		
Appointments	2.64 (1.1–6.6)	<0.05
Medications	5.64 (2.0–15.6)	<0.05
Bills	4.43 (1.6–12.3)	<0.05
Computer	2.39 (0.8–7.5)	0.14
TV	12.6 (1.5–105.1)	<0.05

OR: Odds Ratio, adjusted for age and education. CI: Confidence Interval.

**Table 4 geriatrics-08-00007-t004:** Typology of iADL decline, stratified by referral status.

Number of iADL Domains Showing Decline	iADL Domains	Total (N = 100)	Referred Drivers (N = 50)	Non-Referred Drivers (N = 50)
		n (row %)		
0	No iADL impairment	44	14 (28.0)	30 (60.0)
1	Medication	4	2 (50.0)	2 (50.0)
	Computer	3	1 (33.3)	2 (66.7)
	Appointment	2	0 (0.0)	2 (100)
	Bills	2	1 (50.0)	1 (50.0)
2	Medication, Bills	6	5 (83.3)	1 (16.7)
	Appointment, Medication	5	1 (20.0)	4 (80.0)
	Appointment, Bills	4	2 (50.0)	2 (50.0)
	Appointment, Computer	2	1 (50.0)	1 (50.0)
	Bills, Computer	2	0 (0.0)	2 (100)
	Appointment, TV	1	1 (100)	0 (0.0)
	Medication, Computer	1	1 (100)	0 (0.0)
3	Appointment, Medication, Bills	5	4 (80.0)	1 (20.0)
	Appointment, Medication, TV	3	2 (66.7)	1 (33.3)
	Appointment, Bills, Computer	2	1 (50.0)	1 (50.0)
	Appointment, Computer, TV	2	2 (100)	0 (0.0)
	Appointment, Bills, TV	1	1 (100)	0 (0.0)
	Appointment, Medication, Computer	1	1 (100)	0 (0.0)
	Medication, Bills, TV	1	1 (100)	0 (0.0)
4	Appointment, Medication, Bills, TV	4	4 (100)	0 (0.0)
	Appointment, Medication, Bills, Computer	3	3 (100)	0 (0.0)
	Appointment, Medication, Computer, TV	2	2 (100)	0 (0.0)

## Data Availability

Due to the nature of this study, participants did not consent for their data to be shared or used by external researchers. As such, supporting data is not available.

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
