# Peer review of "Decline in Other Instrumental Activities of Daily Living as Indicators of Driving Risk in Older Adults at an Academic Memory Clinic"

_geriatrics, 2023, doi:10.3390/geriatrics8010007_

Round 1

Reviewer 1 Report (Previous Reviewer 1)

The authors have made many improvements to the paper. The introduction has been revised to be better focused on driving as an iADL and the importance of filling the gaps related to exploring iADL declines in relation to driving risk, and the methods have been revised to better focus on examining declines in iADLs as they relate to driving risk. However, while the authors added a few sentences highlighting the fact that experiencing 3-4 iADL declines seems to be characteristic of those who were referred to the MTO, as opposed to those who were not, more depth needs to be added to the discussion section before the paper can be considered a significant and impactful work. Specifically, the authors need to describe each of the iADLS (appointments, medication, bills, tv, computer) in terms of underlying cognitive functions (aka, what may have declined) and the likely relationship between a decline in that iADL and a decline in driving ability. Additionally, the authors should provide some context for what differentiated older adults who experienced 1-2 iADL declines and were referred to the MTO versus those who were not, perhaps highlighting cognitive, demographic, etc. differences between the groups. Further, the authors should provide details for what mitigating factors were present in older adults who experienced 3 iADL declines yet were not referred to the MTO. This information is vital for physicians, clinicians, and caregivers to better understand which older adults may have a greater chance of referral versus those who may still be safe to drive despite some declines in iADLs.

Author Response

Reviewer 2 Report (Previous Reviewer 2)

I would separate out a section on limitations which is currently included in the discussion section. I might elaborate more on the homogeneity in the study population- ? if data on race/ethnicity, income etc were collected. It could be that these factors could play an additional role in who is referred to the driving authority.  As mentioned in the discussion, referral does not equal actual driving impairment (we don't have the results of any real-world testing or data on accidents etc) so another explanation for some of the findings (like the correlation with age) could be from bias on the part of the referring clinician

Author Response

please see attached. 

Reviewer 3 Report (New Reviewer)

1.- The title is incongruent.

Because in the page 2 of your article, you have written:

"Driving is a complex instrumental Activity of Daily Living (iADL)"

2.- You must say at the beginning, that this is a Secondary Research.  

3.- The sample size is poor.

4.- If you want to do statistical analysis of association as Analytic Study, you must determine a sample size. 

5.- In the table of Number of iADL Functional Impairment.

Total is 100:  R =50,      NR = 50.

With 0  iADL Functional Impairment,  you have total= 44.

R = 14.  It means that the 28% of this group R does not have iADL functional Impairment.  Is it true?.

NR = 30. It means that 60% of this group NR does not have iADL functional Impairment.

6.- The Primary Care Physicians, know to use the :

"Comprehensive Geriatric Assessment" that also includes iADL.

7.- In the bibliography. Did you use Vancouver rules?

The number 14, 17 and 29 are important in this topic.

8.- The number 13: The development of executive function in early Childhood. Monographs of the society for research in child development, i-151.

I think it is other age group.

9.- I recommend to read these articles:

a.- Emily Mitchum, Anne E. Dickerson, PhD.

Relationship Between Instrumental Activities of Daily Living and Naturalistic Driving Performance: Indications for Detection of Mild Cognitive Impairment.

The American Journal of Occupational Therapy 2022 vol 76 (Supplement 1)

b.- David B. Carr

 Driving and the Older Adult (Chapter 51) - Reichel´s Care of Elderly

Published by Cambridge University Press 05 June 2016.

c.- Anne E. Dickerson; Timothy Reistetter; Elin Schold; Miriam Monahan.

Evaluating Driving as a valued Instrumental Activity of Daily Living.

The American Journal of Occupational Therapy 2011, vol. 65(1), 64-75.

d.- Fredrick T Sherman 

Driving: The ultimate IADL.

Geriatrics 2006 October; 61 (10): 9-10. 

e.- Clinician´s Guide to Assessing and Counseling Older Drivers - NHTSA

American Geriatrics Society & A. Pomidor.  Ed (2016, January).

Round 2

Reviewer 3 Report (New Reviewer)

I think that the recommendations that you received and that you have done

in the article, it can be published.

Congratulations !

This manuscript is a resubmission of an earlier submission. The following is a list of the peer review reports and author responses from that submission.

Round 1

Reviewer 1 Report

General Comments

This paper attempts to identify a list of cognitive tests that can be used to help older adults and their caretakers start to discuss driving risk. However, the paper fails to make a case for the unique contribution of this study in the introduction, and the results/discussion confirm that the study has no unique contributions to offer. Instead, the authors determined that the clinical tests that differentiate those who were referred to MTO versus not are the same clinical tests on which the physicians are known to base their decisions at this clinic. Additionally, the discussion section provides no insight into why certain tests or demographic factors might have been significant and in general provides no novel information. I would suggest that the authors significantly revise the introduction to identify what is novel about their study and the discussion to identify and explain some novel findings in these results.

Introduction

Lines 36-42: This paragraph sets up the importance of driving to older adults and that many with dementia have a driver’s license, but it doesn’t provide any clues as to where the study is going. Please add one more sentence to the end of the paragraph stating what the gap in the literature is that this paper is going to address.

Lines 43-52: This paragraph could be better organized to make a stronger point. It starts with driving being an IADL, moves to health and cognition required for driving declining with age, and then reports the different cognitive functions associated with driving. I’d suggest restructuring this paragraph to start with driving being an IADL, listing/explaining the different cognitive functions that support driving, and then identify that these cognitive functions decline with age.

Lines 69-75: This paragraph seems to imply that expert physicians can already pretty reliably identify older adults who should no longer be driving, which means that they likely already have a good list of clinical tests or problematic symptoms that they can use to identify older adults who should no longer be driving. In other words, this paragraph seems to negate the purpose of this paper. Please better identify the unique contribution of this paper. It is currently murky at best.

Data Collection and Analysis

Lines 106-107: This sentence again negates the purpose of the paper. If there’s already a typical set of tests that physicians use to make their decision on driving risk, what is the point of this study? Please better identify the unique contribution of this paper.

Results

Tables: Why are some of the significant p-values explicitly reported (e.g., 0.01 or 0.02) and some are just reported as <.05? I would suggest explicitly reporting all of the p-values.

Lines 182-187: The tests determined to differentiate those referred to the MTO versus not are almost identical to the tests reported in the Data Collection and Analysis section (lines 106-107) as being the ones used by physicians to make their decisions. These results again highlight the fact that this paper does not make a unique contribution to assessing driving risk in older adults.

Discussion

The discussion section is way too brief, with basically one paragraph (lines 206-214) summarizing the results and one paragraph (lines 215-220) relating those results to the literature. More discussion should be included about the meaning of these results and what new information they provide, especially for the 3MS, which is the only reported new finding. Additionally, some discussion should be devoted to the significant demographic factors that differentiated the two groups and why those factors would likely influence driving risk. Finally, because the stated long-term goal was to develop a risk assessment tool in the future, it should be explained how these results could be used in the development of such a tool and how the creation of that tool could be implemented.

Lines 221-222: This limitation is more problematic than the authors make it seem. Because the authors only used data from one clinic and determined that the clinical tests that differentiate those who were referred to MTO versus not are the same clinical tests on which the physicians are known to base their decisions, these results provide no new information. Of course they found strong relationships between the measures the physicians used and the tests that differentiated the two groups of older adults. Had the authors assessed data from multiple clinics and come to the same conclusion, those would have been novel findings.

Reviewer 2 Report

This is well done manuscript. A few non-mandatory suggestions to consider…  As someone who practices in a very diverse environment, it would be helpful to track variables such as race and income level in table 1 as perhaps these variables may also be playing a role in the choice to refer.  I am assuming patients were interviewed in their primary language as fluency can confound testing.  I appreciate that recruitment was consecutive, however by excluding those with incomplete data sets, I wonder if we are are selectively including patients who might have more "warm touches" with the health care system and that this could create another source of bias.

Agree with future directions such as looking whether these variables predict not just referrals but outcomes such as collisions, license revocations etc. Because the largest share of dementia care is likely not done in memory clinics it might be to look at primary care providers without specialized training in geriatrics and whether their referral patterns differ. I suspect these providers are less likely to carry out specialized testing, yet still need to make reasonably standardized decisions on referral.

Author Response

Response to Reviewer:

The authors would like to thank the reviewer for the helpful suggestions to improve our paper. We have detailed our responses below:

Kind regards,

The authors.

Reviewer 2

This is well done manuscript. A few non-mandatory suggestions to consider…  As someone who practices in a very diverse environment, it would be helpful to track variables such as race and income level in table 1 as perhaps these variables may also be playing a role in the choice to refer.  I am assuming patients were interviewed in their primary language as fluency can confound testing.  I appreciate that recruitment was consecutive, however by excluding those with incomplete data sets, I wonder if we are are selectively including patients who might have more "warm touches" with the health care system and that this could create another source of bias.

Thank you for these comments. Unfortunately race and income level were not variables available in this study.

Patients were tested in one of Canada’s two official languages: French and English. We have added a statement to this effect in the methods.

Incomplete data sets were related to inability to do the tests, if the testing created too much stress or if the visits were virtual, excluding some elements. We do not believe there was any bias created by this.

Agree with future directions such as looking whether these variables predict not just referrals but outcomes such as collisions, license revocations etc. Because the largest share of dementia care is likely not done in memory clinics it might be to look at primary care providers without specialized training in geriatrics and whether their referral patterns differ. I suspect these providers are less likely to carry out specialized testing, yet still need to make reasonably standardized decisions on referral.

Thank you for these comments. We have added the idea of iADLs helping primary physicians with decision into the discussion.

Reviewer 3 Report

Review for geriatrics

October 26, 2022

Clinical variables that predict driving risk requiring an older adults referred to an academic memory clinic.

The authors assessed the likelihood of healthcare practitioner reporting to transportation authorities among older adults seen at a tertiary memory clinic, by comparing those who were referred to the Ontario ministry of transportation to those who were not on demographic, health, and certain key clinical variables.  They found that older age, functional impairment, global cognitive impairment, time to complete trails B and clock drawing difficulties predicted such a report.

The article is of some interest, and it could be reformatted as a brief report given the single clinic focus and narrowed set of implications, as long as limitations can be further highlighted.

There are 3 fundamental difficulties with this paper. 

1.       The first is that the authors seem to conflate risk of being reported to risk of driving, which indeed are quite distinct.

2.        The second is that there is a circular logic to the paper–the authors explained that the typical practice of the clinic is to use trails B, the clock and the clinical history to inform whether or not to report.  They then did a statistical analysis that verified that this indeed is the typical practice of the clinic. 

3.       The third concern is that the predictors are not well defined, and inclusion/exclusion criteria are problematic.  For example, in the discussion they mention that 3 or more IADL impairment was a predictor but in the methods and results they mention functional impairment not that particular cutoff.  The authors did not specifically include those with a diagnosis of either mild neurocognitive disorder or major neurocognitive disorder.  Presumably many people attend a memory clinic that do not end up having these diagnoses and including those participants would muddy the data.  Psychiatric diagnosis, cardiovascular diagnosis, was ill-defined.  Were those with significant psychiatric history for example acute psychosis, schizophrenia, active major depression, excluded?  If not, why not?  Were those with post stroke or posttraumatic brain injury excluded if the brain insults were less than 3 months?  6 months?  There did not seem to be a list of exclusion criteria.

Some other challenges, some of lesser significance than above, observed:

1.       The abstract makes reference to "a driving risk tool" that could be helpful.  Indeed this would be helpful but that is outside the scope of what this article and to achieve, and is an example of my concern of conflating different risks.

2.       The conclusion of the abstract simply reiterates the results.

3.       Sentence two of the introduction has no reference.

4.       For line 45 I would reword as “age is the single biggest risk factor for both decline in cognition and dementia” and remove the unfortunately.

5.       Starting on line 51 the references seem to get out of sequence.  For example there is a reference about frontal gray matter that refers to a chapter but then reference #18 seems to address this.  It is unclear why the issue of frontal gray matter is being raised in the introduction when this is not part of the methods or results of the paper.  In line 55-56 I believe reference 18 should be 19.  All of the references need to be double checked including their formatting.

6.       One of the challenges of using neuropsychological testing to predict driving and dementia is the fact that there are no cutoffs.  This should be mentioned in the introduction.

7.       In line 97 it is unclear what the authors mean by medical antecedents.

8.       In 108 the legislation in Ontario was over simplified.

9.       For the not referred group, a 3 MS had to be done in the last 6 months, but for the referred group they had to have a 3 MS, clock drawing test, trails a and trails B in the last 6 months, and it is not certain why there are different requirements for the 2 different groups.

10.   The authors indicate that they dichotomize trails a and B scores to increased statistical power, but that typically reduces rather than increases statistical power.

11.   Late in the manuscript they talked about risk factors of interest, but these are not specified a priori.

12.   It is unclear what demographic factors were adjusted for in the regression analysis in table 3.  Table 1 seems to indicate that there may be important differences in education (again this may have been statistically significant if they had not dichotomized years of education).

13.   The MTO acronym was not defined.

14.   The others did not consider healthcare provider related factors which may predict reporting.

Round 2

Reviewer 1 Report

General Comments

In general, significant changes were made to the results section, but the introduction, methods, discussion, and conclusion sections were left largely unchanged. As they are, the introduction and methods sections focus heavily on cognition, while the results focus heavily on IADLs, causing the paper to be disjointed. The introduction and methods sections need to be significantly revised in order to provide context for and set up the results. The discussion and conclusion also need to be updated to better address the new focus of the study and the new results. A significant rewrite of the entire paper is needed.

Introduction

Lines 63-64: The authors mention “Advanced Driving Directives” without explaining what those are or why they are mentioned. More context for this sentence is needed.

Data Collection and Analysis

The authors did not revise the data collection or analysis, although they stated that they did. There was nothing new about IADLs that was added into the analysis. The current analysis still utilizes the same set of tests as outlined in the previous version of the manuscript to try to differentiate those patients who were referred to the driving authority versus those who were not. This is problematic because the set of tests is almost identical to the tests used by the physicians at the clinic, who used those tests results to determine who was referred to the driving authority. Nothing was changed in the analysis, and thus this paper still does not provide any novel results/information.

Results

The shift in focus in the results is a positive shift towards novel findings.

Discussion

While the authors state that they changed the focus of the paper to IADLs, the authors only added three sentences about IADLs to the discussion, and the discussion of IADLs is still less than half the length of the discussion of the cognitive test results. More discussion should be included about the meaning of these new results and how they relate to the cognitive results. Additionally, some discussion should be devoted to the significant demographic factors that differentiated the two groups and why those factors would likely influence driving risk.

The conclusion needs to be revised to include the new IADL results. It is currently still focused on cognitive test results.

Reviewer 3 Report

Although the authors have reformatted the paper to reflect a focus on iadls, the paper is not of sufficient interest for publication. They do not use a validated iadl measure (or if they do, they do not mention this). The circular argument concern remains even though they’ve de-emphasized this. If the pattern of clinicians is to base decisions on certain cog tests and functional impairment (the latter of which is emphasized in clinical practice guidelines), theyre merely documenting that this clinic is doing a good job following the guidelines. Many subjects have a psychiatric history according to the table and they claim this was likely mild depression on antidepressants but there is no information on how this was ascertained.